# Message in a Bottle: An Exploratory Study on the Role of Wine-Bottle Design in Capturing Consumer Attention

**Emel Ozturk** [1], **Busra Kilic** [1], **Emilia Cubero Dudinskaya** [1], **Simona Naspetti** [2], **Francesco Solfanelli** [1] and **Raffaele Zanoli** [1,*]

1  Department of Agricultural, Food and Environmental Sciences (D3A), Università Politecnica delle Marche, Via Brecce Bianche, 60131 Ancona, Italy

2  Department of Materials, Environmental Sciences and Urban Planning (SIMAU), Università Politecnica delle Marche, Via Brecce Bianche, 60131 Ancona, Italy

*  Correspondence: zanoli@agrecon.univpm.it; Tel.: +39-07-1220-4929

**Abstract:** This study aims to investigate the influence of wine-bottle design and the position of labels on consumers' visual attention in digital contexts. Two within-subjects experiments combined with implicit eye-tracking measures were implemented in Italy. The visual attention of twenty-four participants was measured using areas of interest metrics while being asked to view two differently shaped bottles in three orientations. Subsequently, four examples of each bottle type were displayed, positioning an organic logo in different areas. Attention and interest measures were captured and analysed using a factorial repeated-measures ANOVA. The results show that the shoulder is the bottle's first and most fixated part. Significant differences in participants' attention and interest were found according to the interaction of bottle type and bottle part, as well as bottle part and orientation. Furthermore, exploring the inclusion of an organic logo on different bottle parts provided consistent results. Wine producers and marketers could benefit from bottle anatomy and morphology to identify the best place to display essential information to capture the consumer's visual attention and interest.

**Keywords:** consumer research; packaging design; logo; e-commerce; eye tracking; digital marketing





## 1. Introduction

Online shopping in the European Union (EU) increased from 63% in 2016 to 74% in 2021 among internet users [1,2]. Online offers for wine have also grown [3]. The recent boost in e-commerce activity due to the COVID-19 outbreak pushed wine sales from a 12% increase in 2019 to 43% in 2020 across sixteen key markets, including Italy [4,5]. In general, the online retail wine market is becoming a popular e-commerce niche, which in Italy deserves specific attention, as the country is a leading worldwide producer and consumer of wine [6].

While wine e-commerce is becoming increasingly popular, wine producers can expand into new markets that might otherwise be unreachable [7,8]. Given the nature of e-commerce, consumers cannot experience a wine's intrinsic attributes and sensory qualities [9] and must rely on information provided through the web [10,11]. Nevertheless, the mind needs some quality cues to quickly process the information received during the purchasing process [12]. Consumers rely on visual search cues, such as product and packaging (i.e., bottle and label) characteristics [13,14], as a risk mitigation strategy. In wine e-commerce, this strategy is especially evident among inexperienced or uninformed consumers [15].

Studies focusing on the effects of bottle design on consumers' visual attention are scarce. Although label information, colour, and shape symbolism in the packaging of beverages have been addressed in the literature [16–18], no studies have focused on the effects of bottle morphology and the different physical parts of a bottle on the consumer. Earlier research has indicated the significance of bottle attributes such as shape, size and

colour, as these often help the consumer to identify and categorise the product [19,20]. While previous studies have shown that the orientation of beer bottles can impact consumer attention [21], there is a dearth of research exploring how the bottle morphology and orientation influence consumers' perceptions concerning virtual depictions of wine bottles. Understanding which part of a bottle and what orientation has the most significant effect on consumers' visual attention might help marketers decide to place the most relevant information in that area. Consequently, bottle manufacturers might differentiate their products based on the shape of the bottle part that captures the most consumer attention.

In this context, the aims of the current study were twofold. The main objective was to investigate the role of wine bottles' anatomical and morphological[1] features on consumers' visual attention in digital contexts. The secondary goal was to identify the most prominent bottle parts for labelling in different types of bottles and diverse orientations [21]. To achieve these objectives, the study employs an experimental research design with a repeated measure within-subsect design in combination with an eye-tracking device. Findings can aid in developing more effective marketing strategies for wine manufacturers.

*Conceptual Framework*

Product packaging plays a crucial role in helping consumers assess what kind of product they are about to purchase and its associated quality [22,23]. In the case of wine, bottling is essential for product quality purposes, such as containment, protection, and preservation [24–26], as well as for aesthetic reasons. Indeed, aesthetics increase consumer reaction times and preference over standardised packages in well-known brands, despite price differences [27]. As packaging influences consumer perception and product acceptance [13], wine bottles must be appropriate for use in multiple contexts and as part of specific social rituals [28].

External visual stimuli and package design elements are crucial in capturing consumer attention [22,28,29]. In the literature, packaging design elements are classified into two main categories: visual and informational [30]. While the visual elements include the shape, size and graphics of the packaging, the informational elements consist of the product information from the packaging, e.g., labels. Label information has received great attention in previous research on visual attention as an informational element [28,31–33]. A recent study by Chamorro et al. [34] shows that the label has the biggest influence on consumer choice, followed by bottle type among diverse design elements of a wine bottle. While the front label, where information such as country of origin or brand name is presented, is considered very important as a first communication tool with the consumer, the back label refers to other information such as wine description [35].

In the online retailing context, products are presented as small images of wine bottles showing only the front side, which usually requires the customer to zoom in (or to click to expand and read the text) to read the information on the label [36]. This means that when consumers interact with the different wine bottle offers, the bottle labels are not immediately visible to them. Instead, bottle shape, colour, and size are the most relevant factors affecting consumer choice and attention [19,29,37].

The above framework encompasses most packaging design elements that impact consumer attention within consumer research. Understanding these elements and their impact on consumers can lead to effective design strategies that capture consumer attention in market research. Although bottle design and type in beverage packaging, especially for wine, has previously been explored [34], the role of wine-bottle anatomy and morphology as a consumer's visual attention driver has, until the present moment, mainly been ignored. Utilising the eye-tracking technique to explore consumers' visual attention towards these elements and their interaction can yield valuable insights. This research gap presents an opportunity to investigate the influence of bottle type, bottle part and orientation on consumer attention.

## 2. Materials and Methods

### 2.1. Experimental Design

The study was conducted using a repeated-measures within-subjects experimental procedure in combination with an eye-tracking device. Within-subjects designs combined with eye-tracking devices are frequently used in consumer research on attention as a tool to determine whether visual stimuli are attended to or not [38,39]. In the present study, two within-subjects experiments were conducted. The first experiment, a 2 × 4 × 3 mixed factorial design, measured consumers' attention across two bottle types, four bottle parts and three different bottle orientations. The second experiment, a 2 × 4 design, measured consumer attention on two bottle types and on a logo placed on four different parts of each bottle. In both cases, participants' attention was the dependent variable, captured through an eye-tracking device.

### 2.2. Stimuli

'Bare' bottle images without labels were chosen for the first experiment to avoid any attention bias on the labels or other complex visually processed elements. Since the aim of the study was related to bottle anatomy and morphology, stylised images (silhouettes) of each bottle were used to avoid any reflection, contrast and colour differences coming from the actual photos, which may have impacted the visual attention of the participants. Specifically, two bottle models of the same volume (750 mL) commonly used in the wine-bottling industry but different in design and shape were chosen for this study. The bottles differed in the design (anatomy and morphology) and proportions of their individual parts. While one bottle had a more extended shoulder and shorter body and heel, the other had a shorter shoulder but a longer body and heel (Figure 1).

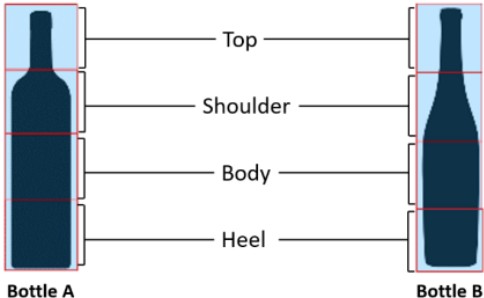

**Figure 1.** Example of the area of interest in the two bottles' silhouettes used in the study.

Each bottle was divided into four equal-sized areas of interest (AOI) (Figure 1). AOIs were defined considering bottle anatomy, and guidelines were suggested according to previous research [40,41]. The four AOIs were defined according to the silhouette of the bottle: (i) top (i.e., the upper part of the bottle); (ii) shoulder; (iii) body (i.e., the central part of the bottle); and (iv) heel (i.e., the lower part of the bottle—the bottom).

For the first experiment, stylised images of each bottle were prepared in three orientations: vertical, horizontal left and horizontal right (Figure 2) to provide more reliable results, as well as presenting the bottle images in different orientations to avoid a bias towards looking at the centre of the viewing area, as mentioned by Tatler [42].

For the second experiment, the European organic logo was placed on each bottle in four different positions, considering the bottles' body parts (Figure 3). The organic logo was selected for two main reasons. On the one hand, the organic logo is a highly valued and trusted label across Europe [43,44], as well as a well-known label component in the wine market [45]. On the other hand, it is mainly composed of visual elements, which consumers tend to prefer over verbal descriptions [28,46,47]. The different bottles' silhouettes, orientations and manipulations were designed using Adobe Photoshop.

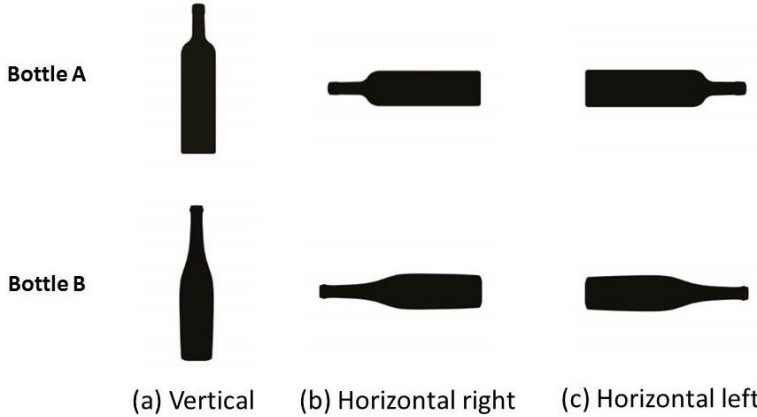

(a) Vertical        (b) Horizontal right        (c) Horizontal left

**Figure 2.** Silhouettes of the two different bottles in three orientations.

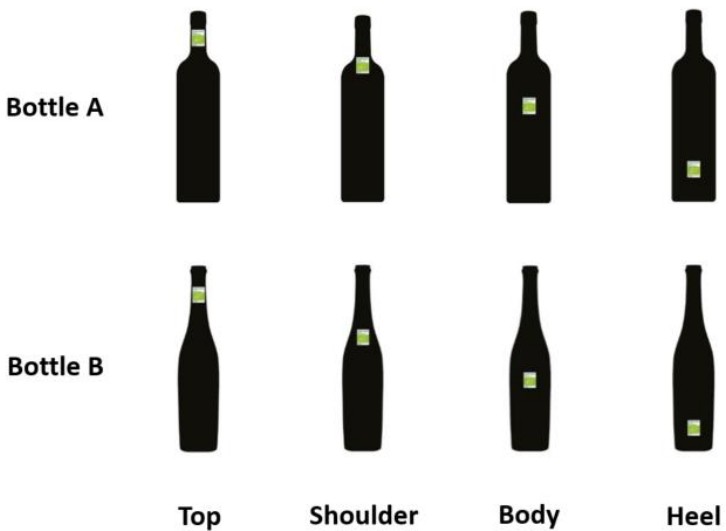

Top          Shoulder          Body          Heel

**Figure 3.** Locations of the organic logo on each bottle type.

### 2.3. Procedure

The experiment took place in a quiet room under standard illumination conditions. Each participant entered the laboratory individually, received the instructions and signed the informed consent form. Participants were seated in front of a monitor and asked to move as little as possible once they had found a comfortable position. The distance from the eye tracker and monitor was respected. In the instructions presented on screen, participants were informed that the survey aimed to identify the most suitable location of the organic production logo for each wine bottle—independently of whether it was situated at the front or back—according to their preferences and the test was started on the screen.

Each step is illustrated in Figure 4. Before data collection, each participant started the experimental session with an eye-tracker calibration task; a nine-point calibration pattern was shown on the screen to allow the eye-tracker to recognise where the person was looking. After calibration, participants were shown a stylised tree image for five seconds. This step aimed to familiarise all participants with a stylised image while also distracting them before being shown the bottle silhouettes. Finally, data collection began, with the participants focusing first on one bottle design and then on the second. The viewing order of the two bottle designs was randomised.



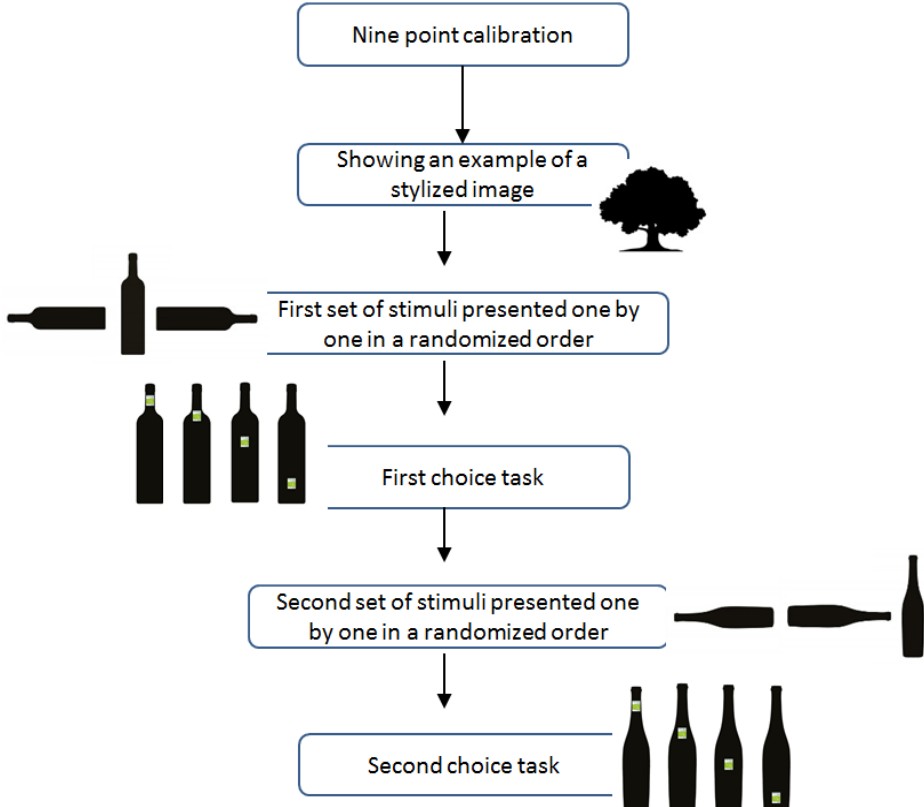

**Figure 4.** Sequence of the task followed by each participant.

For each bottle design in the first experiment, the participants were presented with the silhouettes of the bottle in three different orientations (vertical, horizontal left and horizontal right). Each orientation was shown in a random order according to a Complete counterbalancing for three stimuli [40] to address the sequence effect issue, ensuring that every possible sequence appeared at least once [48]. An exposure time of seven seconds for each image was determined after a pilot test to avoid bias in the resulting time metrics. Each participant observed the bottle in all orientations. The participants were asked to carefully examine each silhouette while the eye-tracking device recorded their fixations. When moving from one silhouette to the next, a random grey inter-slide was presented for five seconds to allow the pupil size to adjust to the same starting value.

After having viewed the various images, each participant was presented with a slide that simultaneously presented four images of the same bottle with the organic logo in a vertical orientation, randomly presented with a focus on different positions (neck, shoulder, body and heel). Participants were asked to select the location they perceived most suitable for the organic logo, according to their preference. Although the choice task provided a consistent setting to study the relevance of the bottle parts in driving the participants' attention, the chosen location was not relevant for this study since the aim was to study the consumer's visual attention when focusing on the individual bottles before performing the actual choice task. In other words, the relevant data collected for this second experiment consisted, once again, of eye-tracking data.

Then, participants repeated the two experimental steps with the second bottle type.

### 2.4. Data Collection and Analysis

The experiment took place at the Food Consumer Research Laboratory of the Marche Polytechnic University (Italy). A convenient sample of students and workers from the Marche Polytechnic University were recruited. All participants were over 18 years of age. The laboratory was equipped with Tobii X2-60 eye tracker compact edition and an LCD monitor with a resolution of 1920 × 1080 (1080P). The eye-tracker was placed unobtrusively

below the computer screen on which stimuli were displayed to the participants. The IMotions® version 7.1 software was used for the design of the experiment, data collection and empirical analysis of eye tracking data.

### 2.4.1. Saliency Controls

As a control analysis, the selected bottles' salient parts were determined by creating their individual saliency maps. Saliency maps compute visually observable locations of a stimulus based on normal visual features that include brightness, colour, and motion. Matlab's graph-based visual saliency model (GBVS) was used to create saliency maps of each bottle. The GBVS is a bottom-up saliency model; according to the literature, it is more accurate in predicting saliency maps than other models [49].

### 2.4.2. Eye Tracking Measures

Mueller and Lockshin [50] concluded that the importance of visual packaging could not be reliably measured using verbal methods. In this respect, eye-tracking technology has gained importance in exploring consumer choices by investigating which elements capture their visual attention. Consumers' visual attention is distributed unconsciously, within seconds [51], tending to favour stimuli unconsciously perceived as being more attractive. The visual properties of stimuli (e.g., colour, darkness, brightness of food packaging) can impact consumers' visual attention [52]. The effect of visual saliency as a precursor of attention in choice situations is well-established in literature: visually salient objects (e.g., labels) capture attention first, and are fixated on first and for a longer time [52–55]. However, greater attention does not automatically translate into choice [52,56,57]. Nevertheless, a visually salient spot is worth further exploitation from marketers by employing cognitively relevant or emotionally arousing stimuli [58].

The data collected with the eye-tracker were first mapped onto each stimulus of bottle silhouettes using heat maps. Heat maps are a colourful representation of how the participants look at the different parts of the stimulus. In the heat maps, the areas that attracted the highest attention are indicated in "red", while those that attracted medium levels of attention are in "yellow". "Green" indicates the areas that attracted the lowest attention. Visual attention is associated with eye fixation and eye movements, and their combination is specific to each individual, indicating the individual's visual attention level [59]. Visual fixation is the most commonly used parameter to determine where consumer attention might be focused [60]. Eye-tracking devices capture eye movements with a frequency of 60 Hz and $-0.4°$ accuracy. Several different measures of eye movements can be obtained with eye tracking, such as the number and duration of fixations, the time spent on a fixation area, and the number of times the gaze returns to this area [61].

The present study focuses on two critical eye-tracking metrics: time to first fixation (TTFF) and time spent (TS). Both measures have been previously used in literature to measure consumers' attention and interest [31,51]. The TTFF measures attention and noticeability [41]. It measures which AOI first attracted the participant's attention. More precisely, it corresponds to the time a participant takes to fixate on a bottle or a bottle part for the first time. Therefore, a shorter TTFF time indicates that the participant fixation started sooner after the image appeared on the screen. A longer TTFF time indicates that the fixation occurred later or did not occur. In the latter case, the TTFF was equal to the maximum time spent on the screen.

On the other hand, TS quantifies the time participants have spent looking at a specific AOI. It represents the participant's interest in the fixated object [40,51]. Higher TS indicates a higher interest from the participant in the fixated object.

Factorial repeated-measures ANOVA procedures were run to determine whether the bottle type (Bottle A vs. Bottle B), bottle parts (top vs. shoulder vs. centre vs. bottom), orientation (vertical vs. horizontal left vs. horizontal right), and interactions affected TTFF or TS. The Greenhouse–Geisser correction was implemented in cases where the sphericity criterion was not met [62–64]. Moreover, a two-way repeated-measures ANOVA procedure

was also estimated to identify the effect of the bottle type and position of the logo within the bottle on consumers' attention (TTFF) and interest (TS). STATA MP version 17 and R Version 4.2.2 [65,66] were used for the statistical analysis. Contrasts and margin estimations were also implemented for a more detailed analysis.

## 3. Results

Twenty-four participants were recruited for the study among university students and non-research staff, with an equal gender distribution. All participants were adults aged between 20 and 59 (the mean age was 30.25 years). The sample size was sufficient to detect at least a 20% difference between the different bottles and AOI within subjects [40]. The results from the study are presented in three sections. First, the saliency maps of the bottle stimuli obtained from the GBVS modelling are introduced. Then, heat maps obtained from eye tracking present a descriptive analysis of the participants' preferences. Finally, the analysis of the AOI metrics obtained from eye-tracking measures is presented.

### 3.1. Saliency Maps Controls

The (predicted) GBVS saliency maps of each stimulus are presented in Figure 5. The maps show how the most salient area for Bottle A refers to the shoulder, where the centre of the shoulder curve is shown in red. In the case of Bottle B, the saliency maps demonstrated that the most salient parts were mainly the top and the heel. However, in the vertical orientation, the neck and the shoulder curve also represented additional slightly salient parts in yellow.

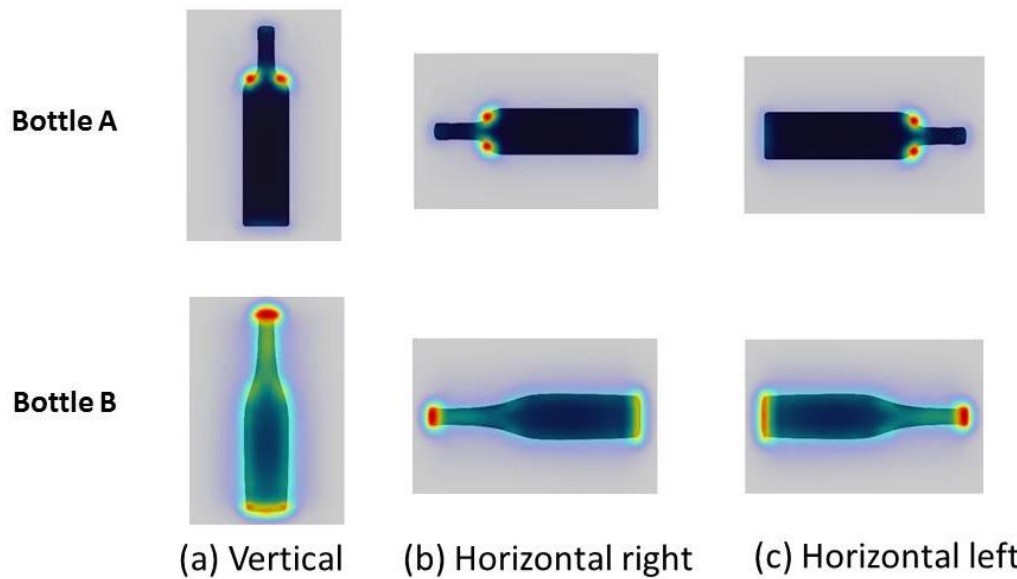

**Figure 5.** Saliency maps of each of the bottles in three orientations.

### 3.2. Heat Maps

The heat maps show that the shoulders of the bottles were the most fixated upon areas regardless of the bottles' orientation and type (Figure 6). However, Bottle A's most fixated areas (red patches) are larger than Bottle B's. Moreover, the second most fixated upon area in Bottle B was the top, regardless of the orientation. These results are consistent with the GBVS saliency maps.

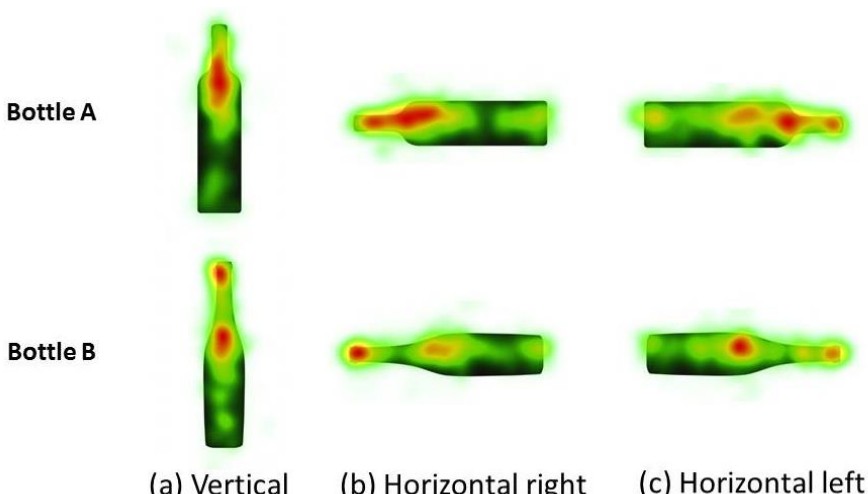

(a) Vertical    (b) Horizontal right    (c) Horizontal left

**Figure 6.** Heat maps of each of the bottles in three orientations showing the areas of fixation combined for all of the participants (red: high; yellow: medium; green: low).

When the silhouettes were presented in a horizontal orientation (both left and right), the heat maps showed that the top was also the second most fixated upon area for Bottle A. Moreover, the bottom of the bottles in a horizontal orientation also obtained the participants' attention. The area that received the least attention in all orientations was the centre of the bottle.

### 3.3. The Role of Bottle Type, Orientation and Bottle Parts

In terms of TTFF, in Bottle A, the shoulder area was the first point towards which respondents directed their attention, followed by the top. This was regardless of the orientation of the bottle. Participants also showed a higher interest (TS) in the same areas (shoulder and top). In the case of bottle B, only the bottles in vertical and horizontal left orientations generated the same visual attention in participants as bottle A (shoulders first, followed by top). However, the participants' interest (TS) differed between the two orientations. Bottle B in the vertical orientation obtained the most interest in the shoulder area with longer TS, while the horizontal left orientation registered it on the top area.

On average, Bottle B in the horizontal right orientation obtained consumers' visual attention in the top and the central areas. The time spent on each part was also among the highest, on average, compared to other bottle parts. Participants looked at the bottom area last. On average, more time was spent fixating on the shoulder area of the bottles than their top, centre and bottom. Detailed results are presented in Table 1.

The factorial repeated-measures ANOVA results showed no significant three-way interaction for TTFF [$F_{(6, 529)} = 1.24$, $p = 0.285$]. However, the bottle type and bottle part interaction had a significant effect on TTFF [$F_{(2.25, 0.75)} = 5.06$, $p = 0.005$ (with Greenhouse–Geisser adjustment)]. In specific, the centre area of the bottles presented a significantly different TTFF [$F_{(1, 568)} = 12.01$, $p = 0.001$] between the bottle types. Although, on average, the participants followed the same fixation pattern for both bottles (first fixated on the shoulder area, followed by top, centre and bottom), the respondents turned their attention to the central area of the bottle faster for Bottle B than for Bottle A. Details can be observed in Figure 7.

**Table 1.** Average means and standard deviations (in parenthesis) of AOIs of each bottle in three positions.

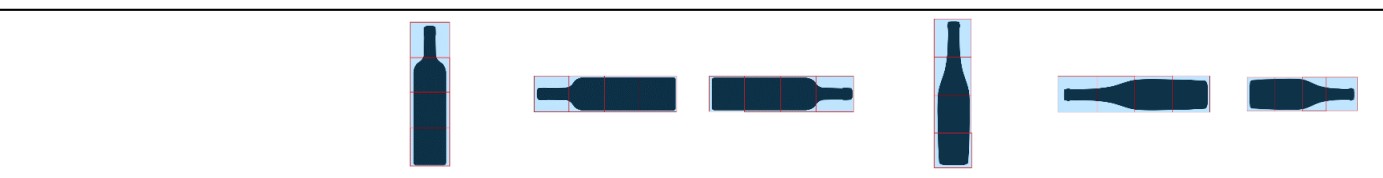

| AOI | AOI Metrics (ms) * | Bottle A | | | Bottle B | | |
|---|---|---|---|---|---|---|---|
| Top | TTFF ** | 2167.46 (2162.33) | 1992.88 (2262.38) | 1913.71 (2188.9) | 1548.08 (1812.17) | 1837.67 (2003.72) | 1582.38 (1662.73) |
| | TS *** | 1228.42 (1132.89) | 1093.46 (905.46) | 1001.17 (834.91) | 1463.04 (1072.06) | 1194.88 (645.78) | 1208.92 (1029.76) |
| Shoulder | TTFF | 548 (1154.67) | 1517.12 (2239.12) | 978.79 (2188.9) | 745.17 (1472.22) | 1445.79 (1722.23) | 2036.08 (2542.84) |
| | TS | 2433.21 (1371.54) | 1655.08 (1194.08) | 1242.42 (781.51) | 1861.21 (1301.35) | 1035.42 (715.84) | 815.38 (716.12) |
| Centre | TTFF | 4085.62 (2953.34) | 3719.46 (2858.58) | 3189.17 (3009.04) | 3922.58 (2524.03) | 1462.62 (1556.62) | 1649.04 (2165.52) |
| | TS | 434 (532.8) | 349.42 (470.77) | 798.79 (930.83) | 616.88 (758.28) | 869.71 (510.77) | 1326.38 (1110.62) |
| Bottom | TTFF | 4166.71 (2566.34) | 3553.79 (2306.3) | 3769.701 (2498.87) | 4671.21 (2695.6) | 3757.96 (2126.59) | 3690.12 (2697.74) |
| | TS | 288.42 (336.2) | 569.5 (577.38) | 457.08 (497.53) | 223.67 (381.9) | 608.67 (755) | 524.46 (537.77) |

Note: Values in parentheses correspond to the standard deviations. * (ms), millisecond; ** TTFF: time to first fixation; *** TS: time spent.

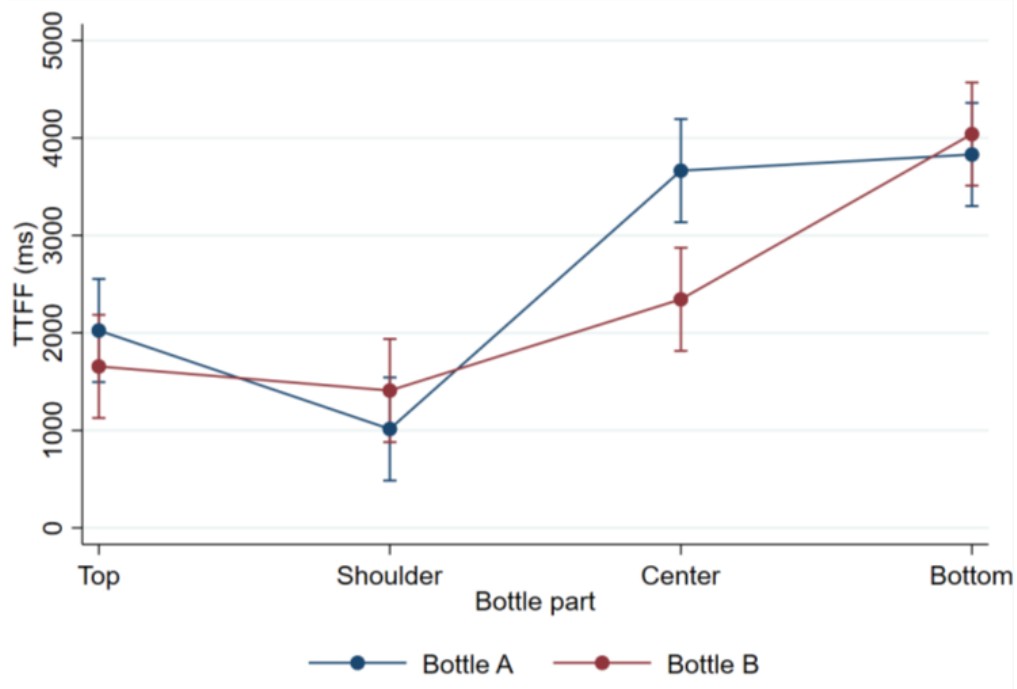

**Figure 7.** Predictive margins for TTFF according to type of bottle and bottle part, with a 95% confidence interval.

The interaction of bottle parts and bottle orientation also significantly affected TTFF [$F_{(6, 529)} = 3.83$, $p = 0.001$]. Although, on average, participants followed the same fixation pattern in all three orientations, significant differences were found for the shoulders and

central areas between the different orientations. Specifically, the TTFF for the bottles' shoulder area in the vertical orientation is about 834.88 ms lower than that in the horizontal left orientation ($p$ = 0.047) and 860.85 ms lower than that in the horizontal right orientation ($p$ = 0.041). This means that participants, on average, fixated sooner on the shoulder area when the bottle was placed in a vertical rather than a horizontal orientation.

Moreover, the TTFF for the bottles' central area in the vertical orientation is approximately 1413.06 ms higher than for the horizontal left orientation ($p$ = 0.001) and 1585 ms higher than for the horizontal right orientation ($p$ < 0.000). In this case, participants fixated on the central part of the bottles oriented horizontally sooner than on the centre of the bottles oriented vertically. Graphical details are presented in Figure 8.

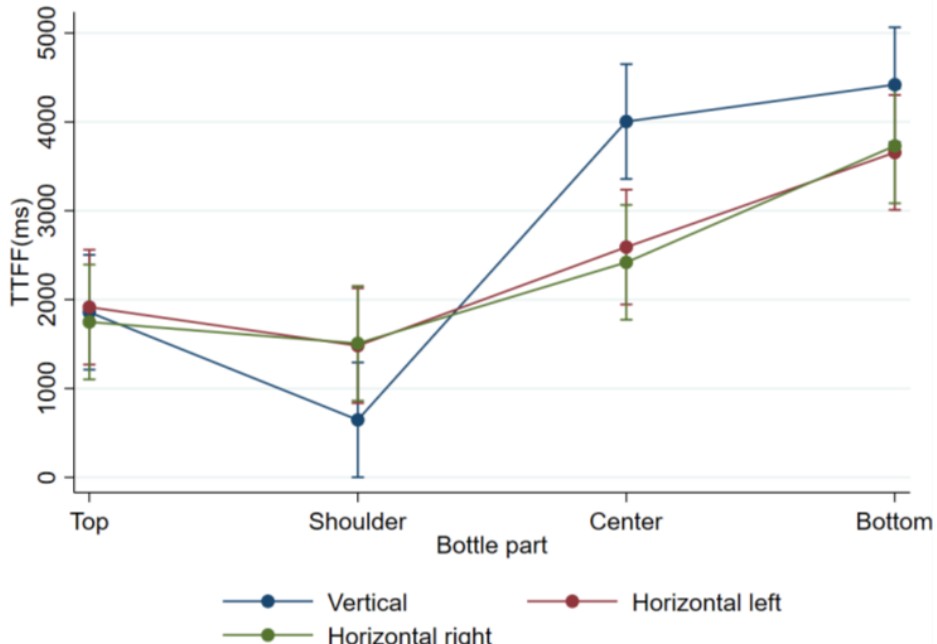

**Figure 8.** Predictive margins for TTFF according to bottle orientation and part, with a 95% confidence interval.

The bottle parts also had a significant main effect on TTFF [$F$(2.15, 0.7174) = 49.99, $p$ < 0.0001 (with Greenhouse–Geisser adjustment)]. Significant effects on the TTFF were found between the different bottle parts [$F$(3, 572) = 39.80, $p$ < 0.0001]. On average, participants fixated on the shoulder of the bottle 628.5 ms before fixating on the top ($p$ = 0.021). Moreover, participants fixated their gaze on the centre of the bottle 1164.39 ms later than on the top ($p$ < 0.0001) and 2094.56 ms later than on the bottom ($p$ < 0.0001).

In the case of time spent, there was no three-way interaction [$F$(2.60, 0.4334) = 0.24, $p$ = 0.8393 (with Greenhouse–Geisser adjustment)]. However, the interaction between bottle type and bottle part had a significant effect on the time participants spent fixating on the selected area [$F$(2.04, 0.6785) = 9.83, $p$ = 0.0001 (with Greenhouse–Geisser adjustment)]. Specifically, there were significant differences between the time spent on the shoulder [$F$(1, 529) = 17.47, $p$ < 0.0001] and central [$F$(1, 529) = 10.10, $p$ = 0.002] areas. On average, participants spent 539.57 ms longer fixating their gaze on the shoulder of Bottle A than on that of Bottle B. Moreover, participants spent 410.25 ms less fixating on the centre of Bottle A than on the centre of Bottle B. A graphical representation is shown in Figure 9.

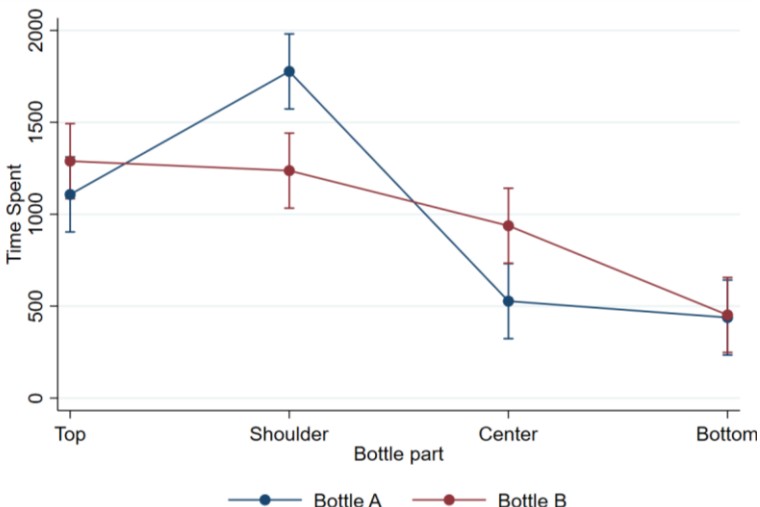

**Figure 9.** Predictive margins for time spent according to type of bottle and bottle part, with a 95% confidence interval.

The interaction of bottle parts and bottle orientation was also significant for TS [$F_{(4.46, 0.7436)}$ = 11.55, $p < 0.0001$ (with Greenhouse–Geisser adjustment)]. Significant differences were found between the time spent on the shoulder [$F_{(2, 529)}$ = 26.59, $p < 0.0001$] and that spent on the centre [$F_{(2, 529)}$ = 6.68, $p = 0.001$]. Specifically, on average, the time spent on the shoulder of the bottle displayed in the vertical orientation was 801.96 ms higher than the time spent in the same bottle part of the bottle displayed in the horizontal left orientation ($p < 0.0001$) and 1118.31 ms higher than the time spent in the same bottle part of the bottle displayed in the horizontal right orientation ($p < 0.0001$).

In the case of the centre of the bottles, there was a significant difference only between the bottles displayed in a vertical orientation and those displayed in a horizontal right orientation ($p = 0.001$). However, on average, participants spent significantly more time fixating on the shoulder area of bottles in vertical orientations than on that of those in horizontal positions [$F_{(2, 115)}$ = 12.89, $p < 0.0001$]. Graphic details are presented in Figure 10.

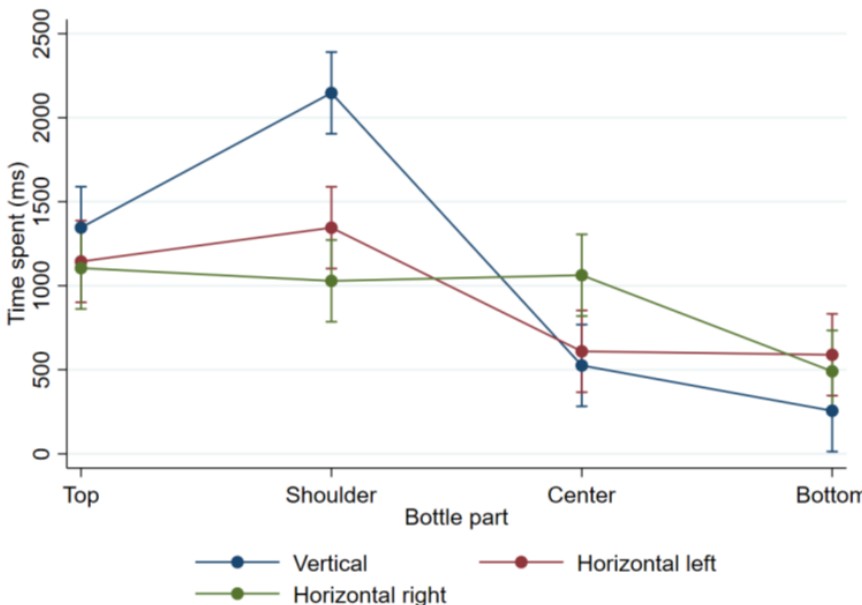

**Figure 10.** Predictive margins for time spent according to bottle orientation and part, with a 95% confidence interval.

The different parts of the bottles also had a significant effect on the time participants spent fixating on them [$F(2.20, 0.7343) = 53.79$, $p < 0.0001$ (with Greenhouse–Geisser adjustment)]. Significant differences were found between the bottle parts [$F(3, 572) = 40.20$, $p < 0.0001$]. On average, participants fixated 308.81 ms longer on the shoulder of the bottle than on the top ($p = 0.004$). Moreover, participants fixated on the centre and the bottom of the bottle, 465.79 ms ($p < 0.0001$) and 753.01 ms ($p < 0.0001$), respectively, less than on the top of the bottle.

### 3.4. The Labelling Effect

When incorporating the label with the organic logo into the vertical bottle silhouettes, there were significant differences in the TTFF for bottle parts with the logo [$F(3, 161) = 2.91$, $p = 0.036$]. Specifically, there are significant differences in the TTFF between the shoulder area and the central [$F(1, 161) = 4.15$, $p = 0.043$] and bottom parts [$F(1, 161) = 8.12$, $p = 0.005$]. On average, participants fixated on the centre and bottom of the bottle 1952.29 ms and 2731.40 ms later, respectively. Regarding the time spent, there were no significant differences according to bottle type [$F(1, 161) = 0.90$, $p = 0.345$], part [$F(3, 161) = 2.45$, $p = 0.065$] or their interaction [$F(3, 161) = 1.53$, $p = 0.208$].

The participants' choice as regards the best bottle part to position the logo was different from the bottle parts they fixated on first or spent more time on. For Bottle A, the preferred location for the logo was the bottom (41.7% of participants), while for Bottle B, it was the central part (33.3%) and the shoulder (29.2%). These choices were also different from the bottles' most salient areas.

## 4. Discussion

The present study investigates the role of the morphology and anatomy of wine bottles on consumers' visual attention in a digital context to identify the most prominent bottle parts for labelling in different types of bottles in diverse orientations. Two within-subjects experimental designs, in combination with an eye-tracking device, were implemented to address the main aim. This experimental design helped avoid automatic biases mentioned in previous studies [42].

The combined fixation data of the participants showed that the participants' attention was initially attracted by the shoulders of the bottles, followed by the tops, the centres and then the bottoms. This pattern was consistent for the bottles and the different orientations in which they were displayed. The different visual saliency of the bottle parts can be the main reason behind these findings. Indeed, the shoulder represents the most relevant shape and salient part in characterising the anatomies of Bottle A and Bottle B (to a lesser degree), as confirmed by the bottle GBVS saliency maps. This is in line with previous research that has concluded that objects with higher saliency are more likely to be fixated on than less salient ones [67–71].

Although there was no significant effect of the type of bottle on the participants' attention and interest, there was a significant effect of the interaction of the type of bottle and the bottle part. In the case of Bottle A, which had a shorter shoulder with sharply cut lines, participants fixated on this area first and spent more time looking at it. In bottle B, which had a softer, smoother shoulder, participants also focused on this area but spent less time on it and instead spent more time fixating on the centre of Bottle B than on that of Bottle A. This finding may be ascribed to the fact that Bottle A has a more pronounced curve in the anatomy of the shoulder, which would be consistent with previous work showing a preference for curved objects with more sides [29,72]. These results suggest that the shoulder area is the best part to focus consumers' visual attention and interest in digital contexts, regardless of the bottle type. However, this is particularly true for bottles with a similar morphology to that of Bottle A.

There was no significant effect on participants' attention or interest according to the bottles' orientation, which is consistent with the results of previous literature [28]. However, there was a significant interaction between the bottles' orientation and part. Participants

fixated first and spent more time on the shoulder area of the bottles in a vertical orientation than on those in a horizontal one, highlighting a possible upper attention bias [73]. However, participants moved their visual attention to the central area more quickly when the bottles were displayed in a horizontal orientation as opposed to when displayed vertically. Such behaviour is consistent with the Western reading habit from left to right, resulting in a higher sensitivity to visual elements on the right field [74]. Therefore, when displayed horizontally, participants would move from one part of the bottle to another more quickly. Wine producers and marketers, when selling online, should consider displaying the bottles in an orientation that motivates consumers' interest and attention to the area in which key information is displayed. For example, if the most determinant labels are displayed at the shoulder of the bottle, the bottle should be presented in a vertical orientation in the online shop.

Adding the organic logos to the bottles' silhouettes provided results for visual attention that were consistent with those obtained from the analysis without the logos. Participants still focused their visual attention first on the logos on the shoulder area and top, followed by those on the centre and bottom. This confirms the importance of the bottles' anatomy and morphology, as visual attention behaviour was similar in both cases, with and without a logo. Moreover, it stresses the fact that the shoulder of the bottle is critical in obtaining consumers' visual attention. There were no significant differences in the time spent on the logos placed on different parts of the bottles. Since the logo was the same, the participants did not require additional processing efforts to understand the logo, regardless of its location.

Concerning the choice of the best location for the organic logo, the results differed between the two types of bottles. While the bottom area was preferred for Bottle A, the centre and shoulder areas were preferred for Bottle B. The selection of these areas was incompatible with the saliency maps or the areas that obtained consumers' visual attention or interest. Although the result might seem surprising initially, previous research has already concluded that greater attention only sometimes translates automatically into choice [52,56,57]. Moreover, top–down factors such as consumer involvement, personal goals, previous knowledge, and familiarity might affect participants' choices [54,75,76]. As these aspects were not included in the study, future research should explore the effect of top–down factors on consumers' choices and weight compared to the characteristics of the stimuli.

These visual attention results can assist in designing wine bottles sold in digital contexts. The present work contributes to the scarce literature on bottle design [19,21] as well as, from a general perspective, on online wine markets [3,10,36,77]. Based on the results of the study, the shoulder area of the bottle receives greater attention from the viewer and is, therefore, potentially relevant as an influence on consumer choice. If these findings can be confirmed in a larger sample, the shoulder areas of bottles can be used to convey important messages about the product the bottles contain. Producers of wine could benefit from placing important certification logos or information in this area to obtain consumers' attention in online contexts.

## 5. Conclusions

The results of this eye-tracking study provide marketers and beverage companies with food for thought as regards optimising their bottles' design. Understanding the role of bottle morphology and anatomy in capturing consumer attention and interest when the wine is sold online might allow them to capture consumer attention and interest better, as the bottle is an intrinsic part of the consumer's first impression of the wine. The results of this study suggest that the different parts that make up the anatomy of a bottle are not equally relevant and that in the labelling and packaging of wine, much more attention should be devoted to the shoulder. Given the small sample size in laboratory conditions, the results of this study need to be confirmed through further research with bigger samples in real-life settings, as well as in other national contexts. Additionally, other studies can further

explore the role of other elements (e.g., label colour and content, bottle colours, etc.) in capturing attention and interest and not just in expectations or associations [78,79]. A final caveat is that this study refers to visual stimuli adapted to wine e-commerce and further eye-tracking studies are suggested in a brick-and-mortar supermarket-shelf environment.

**Author Contributions:** This study was based on the collective effort of the authors. Conceptualisation, E.O. and R.Z.; Data curation, E.O., B.K. and E.C.D.; Investigation, E.O.; Methodology, E.O. and R.Z.; Supervision, R.Z.; Validation, B.K., E.C.D. and S.N.; Writing—original draft, E.O.; Writing—review and editing, B.K., E.C.D., S.N., F.S. and R.Z. All authors have read and agreed to the published version of the manuscript.

**Funding:** This research received no external funding.

**Data Availability Statement:** The dataset is published and accessible at: Ozturk, Emel; Cubero Dudinskaya, Emilia; Kilic, Busra; Mandolesi, Serena; Solfanelli, Francesco; naspetti, simona; Zanoli, Raffaele (2023), "Data for: Ozturk et al. Does the shape and structure of wine bottles influence consumer attention online? An eye-tracking study", Mendeley Data, V2, doi: 10.17632/mhm6zgcsc9.2.

**Conflicts of Interest:** The authors declare no conflict of interest.

## Note

1   For the purpose of this study we refer to 'morphology' in relation to the overall bottle shape, while by 'anatomy' we refer to the shape of the various bottle's parts. Both morphology and anatomy are elements of the bottle's design.

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
