# Peer review of "Message in a Bottle: An Exploratory Study on the Role of Wine-Bottle Design in Capturing Consumer Attention"

_beverages, doi:10.3390/beverages9020036_

Round 1
Reviewer 1 Report
the paper describes the research undertaken, however it is not based in any relevant literature. You need to contextualise the research, in order for it to have meaning.
A second issue is the very small sample size
Author Response
Thank you very much for the time and effort you invested. Please find below a table with your comments and our answer.
|
Reviewers comment |
Authors reply |
|
The paper describes the research undertaken; however, it is not based in any relevant literature. You need to contextualize the research, in order for it to have meaning. |
The context of the paper is to understand the role of bottle anatomy and morphology in attracting consumer attention in the digital context. The contextualization of the research was developed in the introduction. While previous research has studied labels, size, color, etc., bottle morphology has been largely ignored by previous literature. Moreover, the topic is of particular importance in the digital context, where the bottle itself becomes important since consumers interact with a bottle at first sight. This information is provided in detail in lines 53-61. |
|
A second issue is the very small sample size |
While access to a large amount of high-precision data with eye-tracking devices could allow us to use them in statistical analysis, we can also obtain valuable insights with heat maps or a few videos, especially in exploratory research as the current work. In this study, we took as reference Bojko (2013) for the design of the study, in which the sample size was sufficient to detect at least a 20% difference between the different bottles and AOI within subjects (Lines 223-224). Besides, as mentioned by Bojko (2013), within-subject design requires less sample size than between-subject designs. However, we have also mentioned in the conclusion that one of the limitations of the study was the small sample size and it needs to be confirmed by further research with bigger samples (Lines 442-443). |
Reviewer 2 Report
Brief summary:
The paper presents an interesting research on perception of wine packaging. By using the eye tracking technology, the authors investigated how people’s visual attention can be influenced by wine bottle design presented on screen. I think this is a meaningful topic given that online sale is an important part of the winemakers’ success on market nowadays. The article is generally well-written and grounded in relevant literature. Still, I suggest some minor changes:
Specific comments:
1. Line 22: I recommend specifying the type of the label used in the experiment. In general, readers assume that bottle label contains all product-related information (such as brand, product category etc.), but only the organic logo was used in this study. Additionally, I suggest using “organic logo” instead of just “logo” in Line 18.
2. Since the location of the bottle pictures was not manipulated in the study, I recommend changing the name of variable “bottle position” to “bottle orientation” throughout the manuscript.
3. Lines 131-132: It sounds like only two bottles were used. I suggest changing “one bottle” (line 131) and “two bottles” (line 132) to “one bottle design” and “two bottle designs” respectively.
4. Line 141: For how long the grey slide was presented to the participants?
5. Lines 202-203: How the TTFF metric was calculated if a participant did not fixate the AOI at all? In other words, if a fixation did not occur, which value was defined for the case?
6. Line 255: The authors say “bottle B (…) got consumers' visual attention”. Please specify which exact metric was considered here.
7. Lines 270-271: The authors say: ”the centre area of the bottles presented significantly different TTFF”. Did TTFF for the centre significantly differ in relation to all parts (top, shoulder, and bottom) or just to some of them?
8. Lines 280-282, and Lines 335-337: Did statistical analysis include Bonferroni correction?
9. Lines 319-320: The authors say: “participants spent significantly more time fixating on the bottles”. I suggest specifying the area which was fixated on the bottle in this sentence.
10. Lines 359-361: It is worth including more references to research-based evidence which confirm that salient packaging elements influence people’s attention (for example, Kovačević, D.; Brozović, M.; Možina, K. Do prominent warnings make packaging less attractive?, Safety Science 2018, 110, 336-343, doi:10.1016/j.ssci.2018.08.031)
Minor changes:
11. Line 38: It sounds like conjunction word is missing in the sentence, or in-text reference is [9] is oddly placed.
12. The abbreviation AOI should be used uniformly throughout the manuscript, without dots. For example, in Lines 96-98 the authors used A.O.I., which is not in accordance with AOI in Table 1.
13. Line 211: There is an incomplete word in “criterion was not met”
14. Figures 5 and 6 should be larger.
15. In Table 1: To reduce redundancy, I recommend placing (ms) in the column header: AOI metric (ms). Furthermore, if TTFF appears in the table only as abbreviation, the same should be done with TS. Avoid mixing different styles of describing the metrics in the same table.
Overall, the paper is useful and interesting!
Author Response
I am attaching the reply to your useful comments.

Reviewer 3 Report
As such, the research is interesting and applicable in practice.
However, I have a few questions:
Was this an eyetracking study in which research participants looked at physically graspable wine bottles?
If so, why do you state that your outputs are intended for online retailers, or focused on consumer attention in a digital context?
Why, lines 430 and 431, do you state that the results of your study would not hold up in brick-and-mortar stores?
Author Response
Thank you. The pictures were shown online, therefore the study applies only to online retail. I am attaching the reply to your useful comments.

Reviewer 4 Report
The research presents an analysis of the preferences of wine consumers when selecting the purchase through e-commerce.
I consider that the paper does not present results and conclusions that can improve the state of knowledge in the matter studied. From my point of view, the results shown in this study could well be preliminary which may well be the starting point for a work that should be more complete.
The role of wine bottle design in capturing consumer attention must include many more aspects than those studied in this research. Although preferences regarding the shape and the bottle position and the placement of some labels are important, other aspects such as the intrinsic design of the main labels and the back labels, as well as the information provided by them, may be more important than the aspects studied in this research.
I also believe that the paper should have been completed with an essay in which the conclusions obtained by studying bottle selection preferences on e-commerce platforms and with commercial bottles were corroborated. In this “real” test, the track eye technique could be used to determine consumer preferences on commercial bottles, which would allow more useful and applicable conclusions to be obtained for bottle design.
In summary, I consider that the track eye technique shown in this work is very powerful and suitable for analyzing consumer preferences in e-commerce platforms, but it has not been applied to its full potential in this work, therefore providing irrelevant conclusions.
These are the main reasons why I consider that, as the study is proposed, it should be rejected for publication.
Author Response

(The authors gave the same response as above.)

Round 2
Reviewer 1 Report
My original comments still stand, in particular the study lacks a thorough literature review, whioch means the conclusions lack underpinning conceptual support.
Author Response
Response to reviewer 1
Thank you very much for the time and effort you invested. As your last review refers to the original comments, please find below a table with your original comments and our answer. We hope that this time our answers will be satisfactory.
|
Reviewers comment |
Authors reply |
|
The paper describes the research undertaken; however, it is not based in any relevant literature. You need to contextualize the research, in order for it to have meaning. |
The context of the paper is to understand the role of bottle anatomy and morphology in attracting consumer attention in the digital context. The contextualization of the research was developed in a sub-section of the introduction (73-108). |
|
A second issue is the very small sample size |
In this study, we took as reference Bojko (2013)for the design of the study, in which the sample size was sufficient to detect at least a 20% difference between the different bottles and AOI within subjects (Lines 260-261). Besides, as mentioned by Bojko (2013), within-subject design requires less sample size than between-subject design. We have also mentioned in the conclusion that one of the limitations of the study was the small sample size and it needs to be confirmed by further research with bigger samples (Lines 479-481). |
References
Bojko, A. (2013). Eye Tracking the User Experience. Eye Tracking Without Eyes. http://rosenfeldmedia.com/books/eye-tracking/blog/participant-free_ey

Reviewer 4 Report
After reviewing the new version of the manuscript provided by the authors, it can be observed that the main improvements introduced in the text have been primarily focused on correcting some errors in the terminology used, without significant changes being made to address the shortcomings pointed out in my previous review.
------------------------------------
Previous revision
I consider that the paper does not present results and conclusions that can improve the state of knowledge in the matter studied. From my point of view, the results shown in this study could well be preliminary which may well be the starting point for a work that should be more complete.
The role of wine bottle design in capturing consumer attention must include many more aspects than those studied in this research. Although preferences regarding the shape and the bottle position and the placement of some labels are important, other aspects such as the intrinsic design of the main labels and the back labels, as well as the information provided by them, may be more important than the aspects studied in this research.
I also believe that the paper should have been completed with an essay in which the conclusions obtained by studying bottle selection preferences on e-commerce platforms and with commercial bottles were corroborated. In this “real” test, the track eye technique could be used to determine consumer preferences on commercial bottles, which would allow more useful and applicable conclusions to be obtained for bottle design.
In summary, I consider that the track eye technique shown in this work is very powerful and suitable for analyzing consumer preferences in e-commerce platforms, but it has not been applied to its full potential in this work, therefore providing irrelevant conclusions.
------------------------------------
Author Response
Reviewer 4
We want to thank you for your time and effort. As your last review refers to the original comments, please find below a table with your original comments and our answer. We hope that this time our answers will be satisfactory.
|
Reviewers comment |
Authors reply |
|
The research presents an analysis of the preferences of wine consumers when selecting the purchase through e-commerce. I consider that the paper does not present results and conclusions that can improve the state of knowledge in the matter studied. From my point of view, the results shown in this study could well be preliminary which may well be the starting point for a work that should be more complete. |
While we understand that personal opinions might differ, we would like to clarify some aspects related to our contribution to the current state of knowledge in the matter studied. The role of wine-bottle anatomy and morphology as a consumer's visual attention driver has not been studied (Line 45-53, 102-105). Given the development of the e-commerce, the bottle morphology is key in digital contexts, as consumers interact with a bottle at first sight (Pelet et al., 2020), without being able to initially see the label content. This information is provided in lines 93-98. Given the lack of studies on the topic, exploratory studies provide the opportunity to discover something new and interesting, while helpful for theorizing empirical material at an early stage (Swedberg, 2020). Results from exploratory research then are useful for understanding whether the selected research path is worth further developments. In this sense, the current research addresses the gap in consumer attention regarding bottle morphology in digital contexts and pavements the way for future research. Indeed, the results of this study suggest that the different parts that make up the anatomy of a bottle are not equally relevant and that in the labeling and packaging of wine, much more attention should be given to the shoulder. As in any research, this study has also some limitations. They are already mentioned in the Conclusion with suggestions for further studies. |
|
The role of wine bottle design in capturing consumer attention must include many more aspects than those studied in this research. Although preferences regarding the shape and the bottle position and the placement of some labels are important, other aspects such as the intrinsic design of the main labels and the back labels, as well as the information provided by them, may be more important than the aspects studied in this research. |
We agree on the importance of the aspects such as labels and information communicated to the consumer, especially in wine marketing. Indeed, we mentioned in the introduction the previous works on such aspects (Lines 74-92). Our study does not disapprove of the relevance of the labels or other design aspects. Instead, it reports the relevance of where to position labels on the bottles and how to present bottles in the digital commerce of wines which is a lack in previous studies. Adding many more aspects (that have been already studied in previous research) is out of the scope of our work. Moreover, experimental designs as ours require a limited number of manipulations (Bryman, 2012). Including additional aspects will limit the results of the experiment, as it wont be possible to identify the source of the outcome from all the considered aspects. |
|
I also believe that the paper should have been completed with an essay in which the conclusions obtained by studying bottle selection preferences on e-commerce platforms and with commercial bottles were corroborated. In this “real” test, the track eye technique could be used to determine consumer preferences on commercial bottles, which would allow more useful and applicable conclusions to be obtained for bottle design. |
Our study is a laboratory experiment that is commonly used in eye-tracking studies. The main advantage of the lab experiment with eye-tracking is to minimize the irrelevant factors that can influence the measurements. However, it cannot simulate exactly real-world environments (Li et al., 2022). We agree that further studies in real-life contexts with eye-tracking can provide valuable insight into the bottle preferences of consumers. We have added limitations of the laboratory study and suggestions for further studies to the Conclusion (Line 479-481) |
|
In summary, I consider that the track eye technique shown in this work is very powerful and suitable for analyzing consumer preferences in e-commerce platforms, but it has not been applied to its full potential in this work, therefore providing irrelevant conclusions. These are the main reasons why I consider that, as the study is proposed, it should be rejected for publication. |
Thank you for the comment. The research is an exploratory study that addresses the gap in consumer attention regarding bottle morphology in digital contexts. It is the first study focused on these issues in such a context, following a rigorous experimental design using eye-tracking and based on previous research. However, as mentioned in the previous answer, if the reviewer has additional suggestions on how to improve the power of eye-tracking in our study, we are open to hearing these ideas. Nevertheless, as the comment stands right now, it does not provide details on the missing potential. Moreover, we believe that our exploratory study highlights an important gap in the literature and provides an important base for the development of future research. |
Thank you again for your valuable feedback. We highly appreciate the time and effort you invested.
We hope you will find our current version of the paper suitable for publication.
References
Bryman, A. (2012). Social Research Methods (Fourth edi). Oxford University Press.
Li, Y., Liu, B., & Xie, L. (2022). Celebrity endorsement in international destination marketing: Evidence from eye-tracking techniques and laboratory experiments. Journal of Business Research, 150, 553–566. https://doi.org/10.1016/j.jbusres.2022.06.040
Pelet, J. É., Durrieu, F., & Lick, E. (2020). Label design of wines sold online: Effects of perceived authenticity on purchase intentions. Journal of Retailing and Consumer Services, 55, 102087. https://doi.org/10.1016/J.JRETCONSER.2020.102087
Swedberg, R. (2020). Exploratory research. In C. Elman, J. Gerring, & J. Mahoney (Eds.), The Production of Knowledge: Enhancing Progress in Social Science (pp. 17–41). Cambridge University Press.
